# Trend of Polymer Research Related to COVID-19 Pandemic: Bibliometric Analysis

**DOI:** 10.3390/polym14163297

**Published:** 2022-08-12

**Authors:** Williams Chiari, Rizki Damayanti, Harapan Harapan, Kana Puspita, Saiful Saiful, Rahmi Rahmi, Diva Rayyan Rizki, Muhammad Iqhrammullah

**Affiliations:** 1Department of Mathematics, Faculty of Mathematics and Natural Sciences, Universitas Syiah Kuala, Banda Aceh 23111, Indonesia; 2Department of Chemistry, Faculty of Education and Teacher Training, Universitas Serambi Mekkah, Banda Aceh 23249, Indonesia; 3Medical Research Unit, School of Medicine, Universitas Syiah Kuala, Banda Aceh 23111, Indonesia; 4Tropical Disease Centre, School of Medicine, Universitas Syiah Kuala, Banda Aceh 23111, Indonesia; 5Department of Microbiology, School of Medicine, Universitas Syiah Kuala, Banda Aceh 23111, Indonesia; 6Department of Chemistry Education, Faculty of Education and Teacher Training, Universitas Syiah Kuala, Banda Aceh 23111, Indonesia; 7Department of Chemistry, Faculty of Mathematics and Natural Sciences, Universitas Syiah Kuala, Banda Aceh 23111, Indonesia; 8Graduate School of Mathematics and Applied Sciences, Universitas Syiah Kuala, Banda Aceh 23111, Indonesia; 9Department of Life Science and Chemistry, Jacobs University Bremen, 28759 Bremen, Germany

**Keywords:** antiviral, personal protective equipment, plastic pollution, SARS-CoV-2, Scopus, vaccine

## Abstract

Polymeric materials are used for personal protective equipment (PPE), which is mandatory for clinicians to use when handling coronavirus disease 2019 (COVID-19) patients. The development of diagnostic tools and vaccines for severe acute respiratory syndrome coronavirus 2 (SARS-CoV-2) is also dependent on polymer technology. This current report aims to provide readers with the trend of polymer research regarding the COVID-19 pandemic, by employing bibliometric analysis. A literature search on the Scopus database (31 January 2022) was carried out using predetermined terms. Using Scopus database features, the publications were filtered based on the year of publication (2020–2022), types of articles (original research and review), and language (English). The metadata were extracted in a CSV (.csv) file, to be later used in VOSviewer software. The data were presented in a table, graph, and network visualization. As many as 512 publications were included, consisting of 66.4% original research articles and 33.6% review articles. Most of the publications were written by authors whose affiliation was in the United States (*n* = 118, 23%) and covering the Materials Science subject area (*n* = 142, 27.7%). The Ministry of Education of China was the most productive organization, publishing 11 articles. The National Science Foundation of China was the top funding source, supporting 45 publications. Heinz C. Schröder was the most prolific author, publishing nine articles. *Science of the Total Environment* was the leading journal publishing the included studies. The trend of polymer technology related to COVID-19 mostly covers PPE and waste-management themes. The use of polymer technology as a delivery system for the anti-SARS-CoV-2 and COVID-19 vaccine is also among the frequently researched areas. We encourage more research in the field of polymer technology be carried out, to overcome the global pandemic.

## 1. Introduction

Other than threatening human lives [1], the ongoing coronavirus disease 2019 (COVID-19) pandemic has caused problems in multiple sectors including environmental management and the economy [2,3]. COVID-19 has manifested variously in each individual, ranging from asymptomatic or mild symptoms (such as fever, cough, dyspnea, fatigue, and sputum) to life-threatening conditions (such as pneumonia and systemic inflammation) [4,5]. In overcoming this threat, governments around the world have used health promotion to slow down the spread of severe acute respiratory syndrome coronavirus 2 (SARS-CoV-2)—the virus causing COVID-19 [3,6]. A vaccination program has taken place, but its accessibility remains challenging in many countries, especially in developing countries [7]. Repurposed antivirals have been used to inhibit the replication of SARS-CoV-2 upon infection, but they suffer from inconsistent data regarding their efficacy from the trials [8]. Therefore, COVID-19 is still considered a serious threat, suggesting the urge for collaborative contribution from all research fields, including polymer technology.

For the prevention of SARS-CoV-2 transmission, healthcare providers wear personal protective equipment (PPE), including the commonly used one–a face mask, which is made from polymeric materials [9]. Used PPE and plastic waste resulted from the in-hospital management of COVID-19 have been proposed as an adverse impact on the environment [3,10]. Therefore, polymer scientists could contribute to tackling COVID-19 by designing polymeric materials that are both effective at stopping viral transmission and less hazardous to the environment. Moreover, the role of polymer technology research is significant in the development of SARS-CoV-2 diagnostic tools [11,12] and vaccines [13]. Taken altogether, the role of polymer technology in overcoming the pandemic is significant. Herein, we performed a bibliometric analysis on studies covering the theme of polymer technology along with COVID-19, which could help polymer scientists determine the trajectories of their research. Several studies have applauded the use of bibliometric analysis to comprehend the general trend of a specific research topic [14], even in the context of COVID-19 [15].

## 2. Methods

### 2.1. Study Design

This study aimed to obtain insight pertaining to the involvement of polymer technologies in the COVID-19 pandemic by means of bibliometric analysis. The search was conducted by performing a literature search on Scopus database—a widely trusted scientific literature database that allows the search of the literature using basic and advanced methods [16]. The metadata from the published literature were exported from the platform and taken as the input data for bibliometric analysis using VOSviewer software 1.6.17 (Leiden, The Netherland). In this study, no data were used with human or animal involvement, thus, no ethical approval was required.

### 2.2. Search Strategy

The literature search was performed on Scopus database on 31 January 2022, using the following search terms: ((“Coronavirus”, OR “SARS-CoV”, OR “MERS-CoV”, OR “SARS-CoV-2”, OR “2019-nCoV”, OR “COVID-19”, OR “SARS coronavirus”, OR “Wuhan coronavirus”, OR “Severe acute respiratory syndrome coronavirus 2”, OR “SARS coronavirus 2”, OR “COVID”) AND (“polymer”) AND NOT (“polymerase”)). Polymerase was an enzyme catalyzing DNA synthesis and irrelevant to the intended study, hence the exclusion. The search was performed on the title, abstract, and keywords of the published literature. The inclusion and exclusion of the literature have been summarized in a flowchart (Figure 1). Papers should be published from 1 January 2019 to 31 March 2022. No specific exclusion criteria were applied, though the selection of the literature was based on language, so non-English papers were excluded. Metadata including the authors, title, abstract, each author’s affiliation, author’s keywords, journal’s keyword, and journal’s title were extracted and exported as CSV (.csv) files. In the case of disambiguation, cross-checking with the literature and the consensus among authors were conducted.

### 2.3. VOSviewer

The extracted data were then imported to VOSviewer software to construct the network visualization mapping of co-authorship, keywords, and each author’s citation. The bibliometric interpretation was conducted according to the mapping generated by the software.

## 3. Results

### 3.1. Characteristics of Included Papers on Polymer Use in COVID-19 Medical Care

A total of 512 papers consisting of original research articles (*n* = 340, 66.4%) and review articles (*n* = 172, 33.6%) associated with polymer technology and COVID-19 were included. A huge portion of the included reports were published in 2021 (*n* = 344, 67.1%), while less frequent publication was observed in 2020 due to the early rise of the COVID-19 case (Figure 2).

Materials Science (*n* = 165, 32.2%), Chemistry (*n* = 142, 27.7%) and Biochemistry, Genetics, and Molecular Biology (*n* = 128, 25%) were the most-studied subject areas in the related publications (Table 1). *Science of the Total Environment* (*n* = 19, 3.7%) was the leading journal publishing the studies (Table 1). More than a half of the literature was published in the United States (*n* = 118, 23%), China (*n* = 94, 18.4%), and India (*n* = 62, 12.1%) combined (Table 2). The two most productive organizations were the Ministry of Education of China (*n* = 11, 2.1%) and Universitätsmedizin Mainz (*n* = 10, 2%), which were the only two organizations having more than 10 publications during the period. The National Natural Science Foundation of China (*n* = 45, 8.8%), the National Institutes of Health (*n* = 26, 5.1%), and the National Science Foundation (*n* = 23, 4.5%) were the top three funding sources for polymer research related to COVID-19, noting that none of these sources are based in Southeast Asia (Table 2). Finally, the most prolific author by the time the data were retrieved was Schröder, H.C. (*n* = 9, 1.8%), followed by Müller, W.E.G. (*n* = 8, 1.6%) and Neufurth, M. (*n* = 8, 1.6%) (Table 3).

### 3.2. Citations Analysis

The data of the top 10 most-cited studies have been presented (Table 4). Citation evaluation was conducted to analyze the 512 screened documents, and it was found a total of 4293 times, averaging 8.38 citations per paper, with 1431 citations per year (2019–2022). The trend in citations peaked in 2021, noting the fact that the pandemic had just started in 2020. The included studies were cited one time in 2019 (probably a delayed published issue from the journal), 155 times in 2020, 2583 times in 2021, and 1554 times in 2022. It is also worth mentioning that only two studies were cited more than 200 times, two studies were cited more than 100 times, and the others failed to pass the 100-citation mark. The most cited study related to the use of polymer during the COVID-19 pandemic was “COVID-19 face masks: A potential source of microplastic fibers in the environment”, published in *Science of the Total Environment*, authored by Fadare O.O. and Okoffo E.D. in October 2020, for which 249 citations were received since its first publication (83 citations per year since the date of publication).

### 3.3. Co-Authorship Countries

The network visualization and overlay visualization maps of the co-authorship countries on the related studies are presented in Figure 3. A maximum of 25 countries per document and a minimum of five documents of a country were restricted to assess the co-authorship countries on the included studies. After limiting the number of citations of a country to 100, we found 12 out of 80 countries met the threshold. The assessment resulted in recording the number of documents published, total citations, and Total Link Strength (TLS), which determines the number of publications in which two keywords occur together. The United States (documents = 118, citations = 1147, TLS = 50), China (documents = 62, citations = 678, TLS = 42), and India (documents = 94, citations = 958, TLS = 37) were the top three countries, while generally none of the other countries listed were able to pass the 50-published-documents mark.

### 3.4. Co-Occurrence of All Keywords

The network visualization of all keyword co-occurrences is presented in Figure 4. A minimum number of co-occurrences of a keyword was set at five, where 609 out of 7328 keywords meet the threshold. “Human” was the most used keyword (occurrences = 273, TLS = 6891), with “humans” (occurrences = 234, TLS = 6039) and “COVID-19” (occurrences = 269, TLS = 5761) ranked in second and third, respectively. Keywords “nanoparticle” (occurrences = 84, TLS = 2479), “antiviral agent” (occurrences = 51, TLS = 1583), and “vaccines” (occurrences = 33, TLS = 965) are among the most frequent co-occurring keywords after “COVID-19” and “polymer” (along with their synonyms), while non-meaningful keywords (such as human(s), review, animals, and so on) are excluded, even though the occurrences were still lower than 100 times. In the non-overlay visualization (Figure 4a), colors highlighting the keywords indicate their closeness to a research theme. Keywords in red are closely related to protective personal equipment and waste management themes; in blue—sensor or biosensor; and in yellow, green, and purple—medical therapy, vaccine, or drug discovery.

### 3.5. Author Citations

The author citation visualization network has been presented (Figure 5). A minimum number of documents of an author is set at 3, while 15 citations are selected for the minimum number of citations of an author. Of the 2915 authors, 37 meet the threshold. Müller, W.E.G. (documents = 8, citations = 44, TLS = 25) was in a tie with the other four authors of the same TLS for being the most prolific author in the related studies, while it is worth noting that Wang S. (documents = 11, citations = 47, TLS = 25) and Wang X. (documents = 11, citations = 50, TLS = 25) were the only two authors to publish more than 10 studies during the period.

## 4. Discussion

Polymeric materials are used in almost all materials, including those used in medical fields. During the COVID-19 pandemic, the consumption rate of PPE increased, as it is essential in giving protection against SARS-CoV-2 transmission. As we have little knowledge about the virus during the early time of the pandemic, studies have been carried out to comprehend the survivability of SARS-CoV-2 on various surfaces, including that of polymeric material [17]. Simultaneously, researchers were developing polymeric technology that could enhance the protective ability of the PPE [9,18].

However, many have notified the potential detrimental effects of the significantly increased use of PPE to the environment [3,19]. In this present bibliometric analysis, we found that concerns about the plastic pollution threat are mostly shared by the trending papers. The most-cited article found herein discussed the microplastic pollution deriving from COVID-19 face masks [20]. Environmental science is among the top 10 studied subjects, and the journal entitled *Science of the Total Environment* topped the list by publishing polymer-themed articles related to the COVID-19 pandemic. In the network visualization of co-occurrence keywords (Figure 4), PPE and pollution themes formed their own group (indicated in red). This environmental problem should be regarded as urgent, because the effect could occur during the pandemic or later, after the pandemic [21]. Taken altogether, research on polymer topic should not only aim to improve the protective ability of the material, but also to increase its eco-friendliness (i.e., reusability, recyclability, biodegradability, and so on).

The top 10 lists in this study were occupied by subject areas or journals focusing on life sciences and medicine. Keyword “human” or “humans” appeared to be the most co-occurring keywords based on the occurrence and TLS. In addition, “nanoparticle, “antiviral agent”, and “vaccines” were among the most frequently co-occurring keywords. These findings collectively suggest that pharmacological utilities of polymer technology are among the trending research topics during the on-going COVID-19 pandemic.

Polymeric nanoparticles have been investigated for their important role in COVID-19 treatment. Polymeric nanoparticles made by reacting poly(propylene fumarate) and poly(thioketal) with 1,6-hexamethylene diisocyanate, subsequently modified with reactive oxygen species (ROS)-cleavable thioketal diamine, were designed for inflammation-induced acute lung injury [22]. Based on the in vitro and in vivo observations, the material could improve lung damage by downregulating ROS and being antagonistic against neutrophil infiltration and pro-inflammatory proteins in lung tissues [22]. Polymerized-form of proanthocyanidin could possibility inhibit SARS-CoV-2 entry to host cell by inhibiting angiotensin II converting enzyme (ACE2) and viral chymotrypsin-like cysteine protease (3CLpro), though the in vitro neutralization of pseudo-typed SARS-CoV-2 test had a negative result [23]. Polymer-based nanoparticles are notable for their drug-delivery ability, to pass through the epithelial cell’s tight junctions [24] and mucosal tissue [25]. The functionality of polymer-based nanoparticles allows the enhancement of antiviral activities, as suggested by reports investigating anti-human immunodeficiency virus [26] and anti-herpes simplex virus type 1 [27].

Moreover, mRNA vaccine development employs biopolymers (such as poly(L-lysine), DEAE-dextran, polyethylenimine, chitosan, and poly(β-amino esters)) as its carrying system [13]. Even in the development of SARS-CoV-2 immunogen by a research group from Imperial College London, a disulfide-linked poly(amido amine) was investigated for the delivery system [13]. Polymeric nanoparticles, poly(lactic-co-glycolic acid)-polyethyleneimine (PLGA-PEI), were employed as delivery systems for SARS-CoV-2 vaccine adjuvants targeting toll-like receptors (TLRs) and retinoic acid-inducible gene I (RIG-I)-like receptors [28]. Intranasal and intramuscular deliveries of the combination of the adjuvant and PLGA-PEI proved its ability to yield higher spike-protein neutralizing antibody titers in mice [28].

Other than their application for vaccine/adjuvant deliveries and COVID-19 treatment, polymeric materials with nano-architecture could be used as a biosensor. Nanotube polypyrrole-based impedimetric biosensor had been used to monitor anti-SARS-CoV-2 nucleocapsid protein monoclonal antibodies, where excellent COVID-19 immunodiagnostic performance on clinical samples was reported [29]. A robust sensor has been achieved by molecularly imprinted polymer nanoparticles breaking through the challenge of limited temperature and pH ranges possessed by the currently available rapid antigen test [30]. A colorimetric approach had been employed in developing an orange-colored nanoparticle embedded with lactoferrin general capturing agent along with the complementary ACE2-labeled receptor, where the sensor achieved high selectivity toward SARS-CoV-2 and did not respond to MERS-CoV, Flu A, or Flu B contaminant [31].

The ‘antiviral’ keyword also occurred frequently because the polymer-based PPE could be improved by adding antiviral properties. Simple silver nanoparticles coating onto polymers such as polyethyleneimine could yield an antiviral activity with >99.9% SARS-CoV-2 inactivation [32]. A new fabric material with high biodegradability has been reported by a study grafting guanidine-based polymer and neomycin sulfate onto cellulose nonwovens, which successfully achieved virucidal activity >99.35% against SARS-CoV-2 [33]. Such innovation is crucial in overcoming the emerging threat from increased medical waste of PPE. Interestingly, polymers possessing additional antiviral properties could also be utilized as disinfectant. Polyionenes-based disinfectant has been suggested to act as an alternative to the currently available small molecule-based disinfectants that are skin penetrable and possibly harmful to humans, since its SARS-CoV-2 inhibition effectivity reached 99.99% [34].

In this study, we found that the United States is the most productive country in terms of journal publication, which is where 23% of all the published journals are coming from. The United States is only rivaled by China (18.4%) and India (12.1%), while the rest of the top 10 most-productive countries mostly fail to even pass the 10% mark. This is most likely influenced by the amount of funding received for research and publications, as the National Natural Science Foundation of China and the National Science Foundation (United States) are among the top three funding sources, clearly indicating the countries’ intentions to share concern about polymer technology for tackling the COVID-19 pandemic. However, it is noteworthy that funding information provided from the Scopus database as used herein could have some errors [35], hence requiring further analysis for the judicious interpretation of this finding. The Ministry of Education China along with Universitätsmedizin Mainz produces a total of 4.1% of the total publications, becoming the most productive organization compared to the others.

Publications on polymer research related to COVID-19 reached 512 articles, where 66.4% are research articles, and the rest are review articles, suggesting the predominance of experimental work. However, this number is still minimal in comparison to the total COVID-19 studies included in the Scopus database reaching 218,564 articles (searched using the same terms combination of COVID-19 and its synonyms along with the same exclusion method herein). Furthermore, cooperation among authors is not as massive as other common bibliometric findings on different topics [36]. Wang is the only author among the top 10 most prolific authors who had collaboration with other authors.

This study has several limitations in which the data are retrieved from one database, two keyword terms are used, and search refinements are based on year of publication, document type, and language, which resulted in the elimination of illegible documents. We only used Scopus, though based on a report published in 2020, this database has poorer accuracy for funding information in comparison to other databases such as the Web of Science [35]. Moreover, the COVID-19 pandemic started only less than three years ago and is still ongoing, hence, the data included herein were relatively small.

## 5. Conclusions

The United States (23%) and China (18.4%) are the most prolific countries in terms of polymer research related to COVID-19. The most prolific author was Heinz C. Schröder (from the University Medical Center of the Johannes Gutenberg University), and the leading journal was *Science of the Total Environment*. Waste management was revealed as the most reported topic, indicating the importance of the development of sustainable material that could ease the waste burden from the COVID-19 pandemic. The combination of polymer technology and COVID-19 as a research focus still suffers from scarcity. Cooperation among researchers from the polymer science field also needs to be strengthened further to overcome the current pandemic and future pandemics.

## Figures and Tables

**Figure 1 polymers-14-03297-f001:**
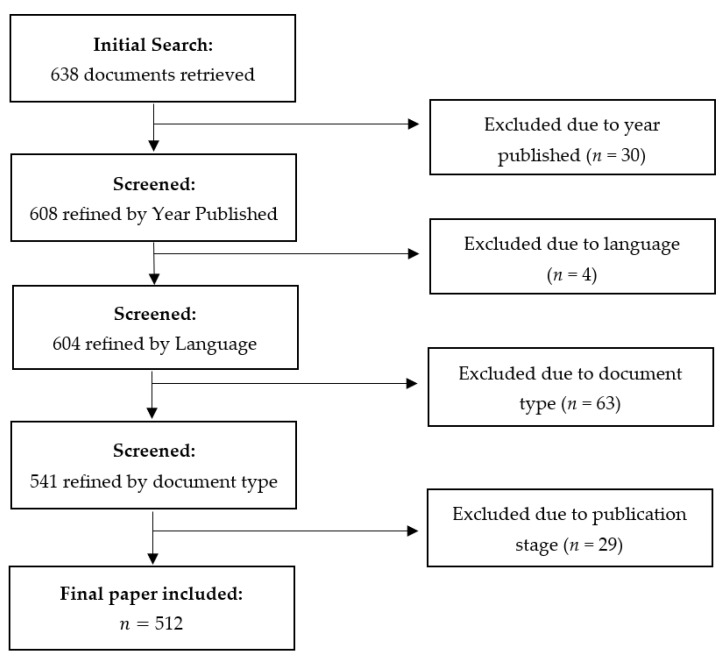
Flowchart of included studies reporting on polymers during the COVID-19 pandemic.

**Figure 2 polymers-14-03297-f002:**
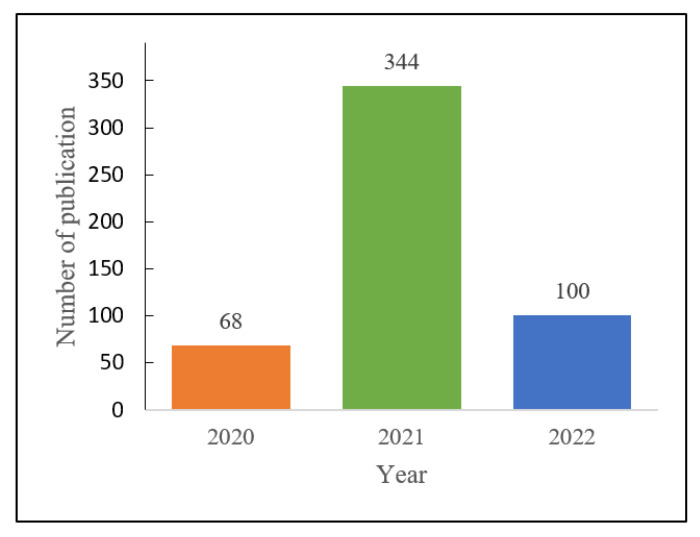
Publication trends of polymer research related to COVID-19.

**Figure 3 polymers-14-03297-f003:**
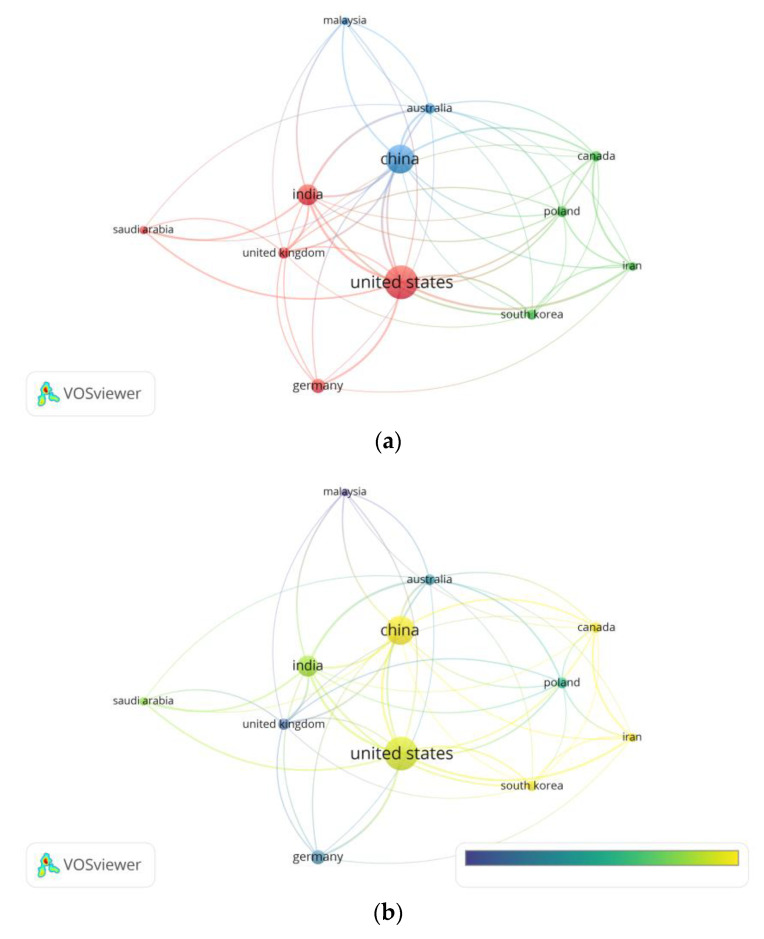
(**a**) Network visualization of co-authorship countries (weights: documents). (**b**) Overlay visualization of co-authorship countries in 2019–2022 (weights: documents; scores: average publications per year).

**Figure 4 polymers-14-03297-f004:**
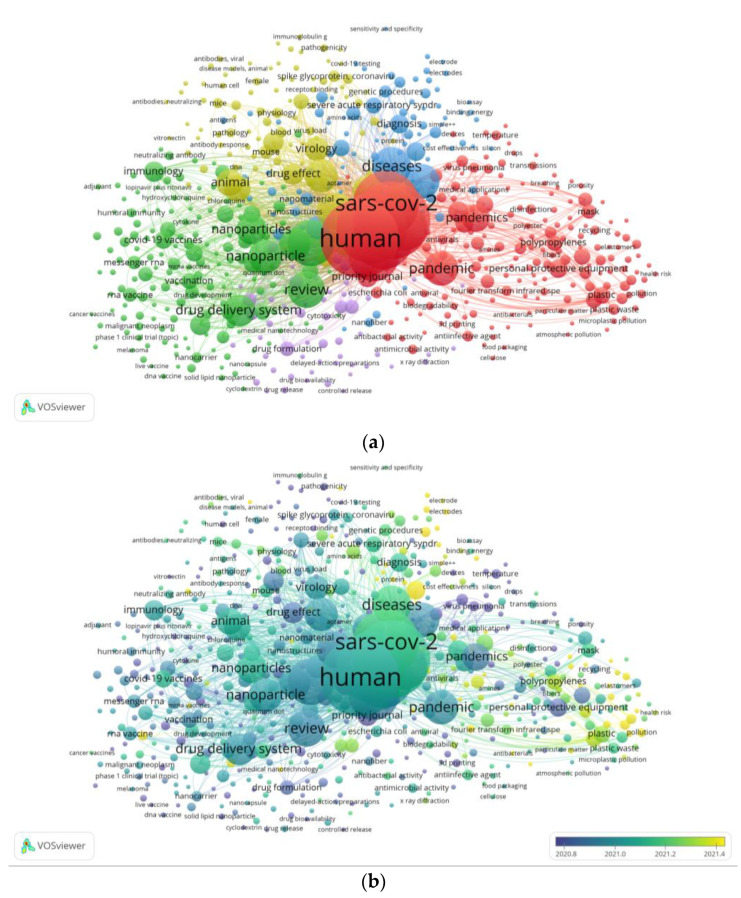
(**a**) Network visualization of all keywords (weights: occurrences). (**b**) Overlay visualization of all keywords in 2019–2022 (weights: occurrences; score: average publications per year).

**Figure 5 polymers-14-03297-f005:**
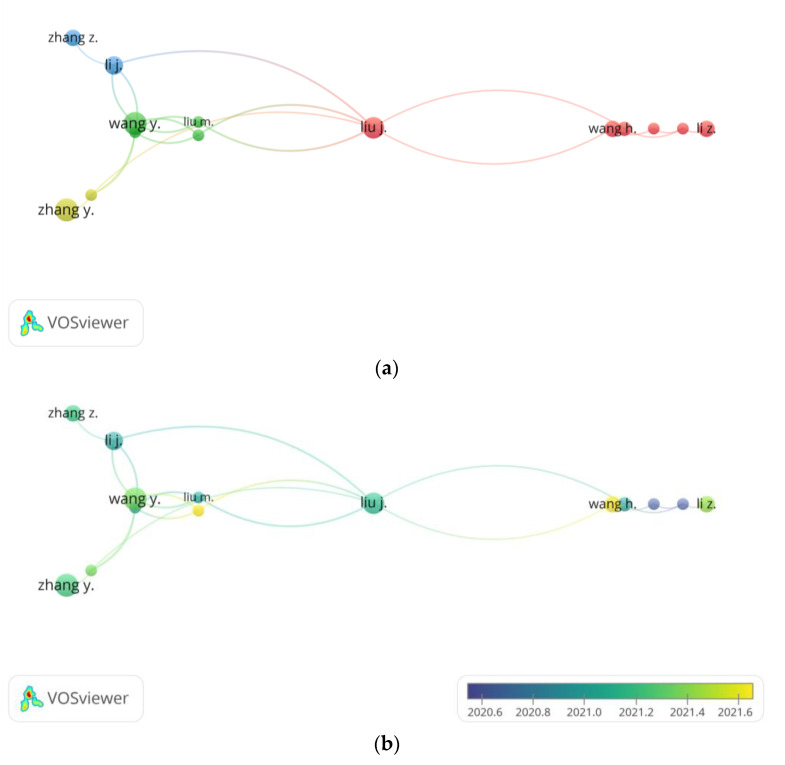
(**a**) Network visualization of authors citation (weights: citations). (**b**) Overlay visualization of author citations in 2020–2021 (weights: citations; score: average publications per year).

**Table 1 polymers-14-03297-t001:** Top 10 subject areas and journals of the polymer research related to COVID-19.

Characteristic	Publications	Percentage (%)
**Subject area**		
Materials Science	165	32.2
Chemistry	142	27.7
Biochemistry, Genetics, and Molecular Biology	128	25.0
Engineering	117	22.9
Chemical Engineering	109	21.3
Pharmacology, Toxicology, and Pharmaceutics	98	19.1
Medicine	91	17.8
Environmental Science	79	15.4
Physics and Astronomy	58	11.3
Immunology and Microbiology	31	6.1
**Journal**		
*Science of the Total Environment*	19	3.7
*Polymers*	15	2.9
*International Journal of Biological Macromolecules*	11	2.1
*International Journal of Molecular Sciences*	10	2.0
*International Journal of Pharmaceutics*	9	1.8
*Journal of Hazardous Materials*	8	1.6
*ACS Applied Materials And Interfaces*	7	1.4
*Pharmaceutics*	7	1.4
*Advanced Healthcare Materials*	5	1.0
*Advanced Materials*	5	1.0

**Table 2 polymers-14-03297-t002:** Top 10 most prolific countries, productive organizations, and funding sources of polymer research related to COVID-19.

Characteristic	Publications	Percentage (%)
**Most prolific country**		
United States	118	23.0
China	94	18.4
India	62	12.1
Germany	35	6.8
Italy	31	6.1
United Kingdom	27	5.3
Australia	24	4.7
Poland	24	4.7
Canada	23	4.5
South Korea	23	4.5
**Most productive organization**		
Ministry of Education China	11	2.1
Universitätsmedizin Mainz	10	2.0
CNRS Centre National de la Recherche Scientifique	8	1.6
Chinese Academy of Sciences	7	1.4
Johannes Gutenberg-Universität Mainz	7	1.4
Queensland University of Technology	6	1.2
Northeastern University	6	1.2
National University of Singapore	5	1.0
New Jersey University of Technology	5	1.0
University of Technology Sydney	5	1.0
**Funding sources**		
National Natural Science Foundation of China	45	8.8
National Institutes of Health	26	5.1
National Science Foundation	23	4.5
National Research Foundation of Korea	18	3.5
European Commission	15	2.9
Natural Science and Engineering Research Council of Canada	13	2.5
Conselho Nacional de Desenvolvimento Científico e Tecnológico	12	2.3
Coordenação de Aperfeiçoamento de Pessoal de Nível Superior	12	2.3
Department of Science and Technology, Ministry of Science and Technology, India	10	2.0
Engineering and Physical Sciences Research Council	10	2.0

**Table 3 polymers-14-03297-t003:** Top 10 authors of polymer research related to COVID-19.

Name	Affiliation	Publication	Percentage (%)
Heinz C. Schröder	University Medical Center of the Johannes Gutenberg University	9	1.8
Werner E.G. Müller	University Medical Center of the Johannes Gutenberg University	8	1.6
Meik Neufurth	University Medical Center of the Johannes Gutenberg University	8	1.6
Xiaohong Wang	University Medical Center of the Johannes Gutenberg University	8	1.6
Shuang Wang	Shenyang Pharmaceutical University	7	1.4
Chaudhery M. Hussain	New Jersey Institute of Technology	4	0.8
Cameron Alexander	University of Nottingham, Queen’s Medical Centre	3	0.6
Elham Azadi	Isfahan University of Technology	3	0.6
Anna K. Blakney	University of British Columbia	3	0.6
Jin-Ho Choy	Dankook University	3	0.6

**Table 4 polymers-14-03297-t004:** Top 10 most cited studies on polymer technology research related to COVID-19.

#	Title	Author(s)	Journal	Year	Citation(s)	Average Citations Per Year	Citations as of 2022
1	COVID-19 face masks: A potential source of microplastic fibers in the environment	Fadare O.O. and Okoffo E.D.	*Science of the Total* *Environment*	2020	249	83	74
2	The effect of temperature on persistence of SARS-CoV-2 on common surfaces	Riddell et al.	*Virology Journal*	2020	202	67.3	34
3	Electrochemical biosensors for pathogen detection	Cesewki E. and Johnson B.N.	*Biosensors and* *Bioelectrics*	2020	180	60	38
4	Challenges, opportunities, and innovations for effective solid waste management during and post COVID-19 pandemic	Sharma et al.	*Resources, Conservation and Recycling*	2020	179	59.6	53
5	Nanomaterial delivery systems for mRNA vaccines	Buschmann et al.	*Vaccines*	2021	98	49	29
6	Optimizing use of theranostic nanoparticles as a life-saving strategy for treating COVID-19 patients	Itani et al.	*Theranostics*	2020	76	25.3	9
7	Flexible nanoporous template for the design and development of reusable anti-COVID-19 hydrophobic face masks	El-Atab et al.	*ACS Nano*	2020	73	24.3	10
8	The SARS-CoV-2 nucleocapsid protein is dynamic, disordered, and phase separates with RNA	Cubuk et al.	*Nature Communications*	2021	72	36	26
9	Antiviral potential of nanoparticles—can nanoparticles fight against coronaviruses?	Gurunathan et al.	*Nanomaterials*	2020	64	21.3	14
10	Methods of inactivation of SARS-CoV-2 for downstream biological assays	Patterson et al.	*Journal of Infectious* *Disease*	2020	63	21	16

## Data Availability

The underlying data of this study can be requested from the corresponding author on a case-by-case basis.

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
