# Peer review of "Trend of Polymer Research Related to COVID-19 Pandemic: Bibliometric Analysis"

_polymers, 2022, doi:10.3390/polym14163297_

Round 1
Reviewer 1 Report
Some suggestions for “Trend of polymer research related to COVID-19 pandemic: Bibliometric analysis”
1. The authors should explain more about the search terms used in this study. Why “polymerase” should be excluded?
2. Some previous studies have doubted the “Accuracy of funding information in Scopus”, please note this point in related analysis and add a note for readers.
3. Line 187: what does “Journal of Infectious Disease0073” mean?
4. Be careful about the various keywords provided in Scopus.
5. How do the authors solve the name disambiguation problem?
Author Response
Dear Reviewer 1,
I thank you for your valuable time checking our manuscript. Below are our responses:
- The authors should explain more about the search terms used in this study. Why “polymerase” should be excluded?
Response: Thank you for this necessary disclaimer to make sure the protocol is well justified. We have included the explanation in Line 103-105.
“Polymerase was an enzyme catalyzing DNA synthesis and irrelevant to the intended study, hence the exclusion.”
- Some previous studies have doubted the “Accuracy of funding information in Scopus”, please note this point in related analysis and add a note for readers.
Response: Thank you very much for notifying regarding the reliability of Scopus database for funding information. To make more judicious interpretation by readers we have incorporated the disclaimer in the discussion (Lines 360—363 and 379—381).
Line 360—363:
“However, it is noteworthy that funding information provided from the Scopus database as used herein could have some errors [35], hence requiring further analysis for judicious interpretation of this finding.”
Line 379—381 (limitation):
“We only used Scopus, in which based on a report published in 2020, this database has poorer accuracy for funding information in comparison to other database such as Web of Science [35].”
- Line 187: what does “Journal of Infectious Disease0073” mean?
Response: Our apology for the typo, we have revised it: “Journal of Infectious Disease”
- Be careful about the various keywords provided in Scopus. And 5. How do the authors solve the name disambiguation problem?
Response: Thank you for your concerns. We respond point 4 and 5 at the same time. For the keywords, we have used synonyms and tested several times to see if the included literatures are the proper one. As for the case of disambiguation, we have performed cross-checking and discussion with all authors when necessary.
In the text we have added this statement Line 110—111:
“In the case of disambiguation, cross-checking to the literatures and consensus among authors were conducted”.
Reviewer 2 Report
Comments on polymers-1863476
In this manuscript, the authors employed bibliometric analyses to study the trend of polymer research regarding COVID-19. The manuscript was written well, and the presentation of data was clear. The authors need to address a few minor points before publication:
1. The information presented in Table 1 was not organized well and thus it was hard to read. The content should be divided into several tables instead of one.
2. For “Top 10 authors” in Table 1, the organizations of the authors should be provided.
3. The authors are encouraged to analyze the most common applications or materials that were reported related to polymer research for COVID-19.
4. The first letter of the countries’ names shown in Figure 3 should be capitalized.
Author Response
Dear Reviewer 2,
Thank you for taking your time evaluating our manuscript. Please find below our responses:
- The information presented in Table 1 was not organized well and thus it was hard to read. The content should be divided into several tables instead of one.
Response: Thank you for this suggestion to improve the readability of our data. We have divided the table into three, please refer to Table 1,2, and 3.
- For “Top 10 authors” in Table 1, the organizations of the authors should be provided.
Response: Thank you for this suggestion to improve the readability of our data. Their affiliations have been added (Table 3). We also think the full names are important, hence we made the change as well.
- The authors are encouraged to analyze the most common applications or materials that were reported related to polymer research for COVID-19.
Response: Thank you for recognizing what is lacking in our discussion. We have added more discussion about the applications of the polymer technology. Please refer to Line 291—344.
“Polymeric nanoparticles have been investigated for their important role in COVID-19 treatment. Polymeric nanoparticles made by reacting poly(propylene fumarate) and poly(thioketal) with 1,6-hexamethylene diisocyanate, and subsequently modified with reactive oxygen species (ROS)-cleavable thioketal diamine were designed for inflammation-induced acute lung injury [22]. Based on the in-vitro and in-vivo observa-tions, the material could improve lung damage by downregulating ROS and being antagonistic against neutrophil infiltration and pro-inflammatory proteins in lung tissues [22]. Polymerized-form of proantho-cyanidin could possibility inhibit SARS-CoV-2 entry to host cell by inhibit-ing angiotensin II converting enzyme (ACE2) and viral chymotrypsin-like cysteine protease (3CLpro), though the in vitro neutralization of pseudo-typed SARS-CoV-2 test had a negative result [23]. Polymer-based nanopar-ticles are notable for their drug delivery ability to pass through the epithe-lial cell tight junctions [24] and mucosal tissue [25]. The functionality of polymer-based nanoparticles allows the enhancement of antiviral activi-ties, as suggested by reports investigating anti-human immunodeficiency virus [26] and anti-herpes simplex virus type 1 [27].
Moreover, mRNA vaccine development employs biopolymers (such as poly(L-lysine), DEAE-dextran, polyethylenimine, chitosan, and poly(β-amino esters)) as its carrying system [13]. Even in the development of SARS-CoV-2 immunogen by a research group from Imperial College Lon-don, a disulfide-linked poly(amido amine) was investigated for the deliv-ery system [13]. Polymeric nanoparticles, poly(lactic-co-glycolic acid)-polyethyleneimine (PLGA-PEI), were employed as delivery very systems for SARS-CoV-2 vaccine adjuvants targeting toll-like receptors (TLRs) and retinoic acid-inducible gene I (RIG-I)-like receptors [28]. Intranasal and in-tramuscular deliveries of the combination of the adjuvant and PLGA-PEI proved its ability to yield higher spike-protein neutralizing antibody titers in mice [28].
Other than their application for vaccine/adjuvant deliveries and COVID-19 treatment, polymeric materials with nano-architecture could be used as biosensor. Nanotube polypyrrole-based impedimetric biosensor had been used to monitor anti-SARS-CoV-2 nucleocapsid protein monoclo-nal antibodies, where excellent COVID-19 immunodiagnostic performance on clinical samples was reported [29]. A robust sensor has been achieved by molecularly imprinted polymer nanoparticles breaking through the challenge of limited temperature and pH ranges possessed by the currently available rapid antigen test [30]. A colorimetric approach had been em-ployed in developing an orange-colored nanoparticle embedded with lac-toferrin general capturing agent along with the complementary ACE2-labeled receptor, where the sensor achieved high selectivity toward SARS-CoV-2 and did not response to MERS-CoV, Flu A, or Flu B contaminant [31].
‘Antiviral’ keyword also occurred frequently because the polymer-based PPE could be improved by adding antiviral properties. Simple silver nanoparticles coating onto polymers such as polyethyleneimine could yield an antiviral activity with >99.9% SARS-CoV-2 inactivation [32]. A new fab-ric material with high biodegradability has been reported by a study graft-ing guanidine-based polymer and neomycin sulfate onto cellulose nonwovens that successfully achieved virucidal activity >99.35% against SARS-CoV-2 [33]. Such innovation is crucial in overcoming the emerging threat from increased medical waste of PPE. Interestingly polymers pos-sessing additional antiviral properties could also be utilized as disinfectant. Polyionenes-based disinfectant has been suggested to act as an alternative of currently available small molecule-based disinfectants that are skin pen-etrable and possibly harmful to human, where its SARS-CoV-2 inhibition effectivity reached 99.99% [34].”
- The first letter of the countries’ names shown in Figure 3 should be capitalized.
Response: Our deep apology for not being able to fulfill this request. As much as we want to improve the presentation of Figure 3, unfortunately the feature of VoSviewer is still limited for doing so. Some literatures, I believe, also have the same problem of not being able to present country’s name with initial capital (for example, doi: 10.1016/j.clet.2022.100437). Perhaps the reviewer would have a suggestion?
Reviewer 3 Report
The manuscript, entitled "Trend of polymer research related to COVID-19 pan- 2 demic: Bibliometric analysis", is a very well written one, clearing reviewing the polymer research trend/statistics related to Covid in the past three years. The authors utilized searching tools, and visualization tools, to clearly reveal status of research in this field. I could see the impact of this work on the future research directions. I would recommend the publication of this manuscript.
Author Response
We thank you for your encouraging words!
Round 2
Reviewer 1 Report
1. For the search query, some references should be given.
2. The search query should be refined. Since you have used the keyword “Coronavirus”, why phrases such as "SARS/Wuhan coronavirus" should be used again?
3. It seems the keyword “polymer” will not necessarily retrieve all results related to “polymerase”.
4. In the keywords analysis section, I think the keywords such as nanoparticles and nanoparticle should be merged.
5. For the use of Scopus in this study, previous studies have already showed that both Scopus and Web of Science are widely used in academic papers. Some others reasons such as database availability are also acceptable.
6. Some well-known studies focusing on the bibliometric analysis of COVID-19 from the Scientometric community should be acknowledged.